# Exploring the role of image domains in self-supervised DNN models of rodent brains

**Aaditya Prasad**
University of California, San Diego
Salk Institute for Biological Studies
La Jolla, CA 92093
`aprasad@salk.edu`

**Uri Manor**
Salk Institute for Biological Studies
La Jolla, CA 92093
`umanor@salk.edu`

**Talmo Pereira**
Salk Institute for Biological Studies
La Jolla, CA 92093
`talmo@salk.edu`

## Abstract

Biological visual systems have evolved around the efficient coding of natural image statistics in order to support recognition of complex visual patterns. Recent work has shown that deep neural networks are able to learn similar representations to those measured in visual areas in animals, suggesting they may serve as models for the brain. Varying the network architecture and loss function has been shown to modulate the biological similarity learned representations, however the extent to which this results from exposure to natural image statistics during training has not been fully characterized. Here, we use self-supervised learning to train neural network models across a range of data domains with different image statistics and evaluate the similarity of the learned representations to neural activity of the mouse visual cortex. We find that networks trained on different domains also exhibit different responses when shown held-out natural images. Furthermore, we find that the degree of biological similarity of the representations generally increases as a function of the naturalness of the data domain used for training. Our results provide evidence for the idea that the training data domain is an important component when modeling the visual system using deep neural networks.

## 1 Introduction

In recent years, it has been shown that task-driven deep neural networks (DNNs) are highly accurate normative models for predicting the neural responses in both the primate [22, 21] and mouse ventral visual stream [2, 15, 10, 4]. Despite their success as models of the brain, it has not been fully elucidated how each component of the DNN contributes to its capacity to learn biologically similar representations. Namely, three aspects of DNNs may be essential: the **network architecture**, the **loss function**, and the **training data**.

Recent studies argue that the **network architecture** can have a significant impact on biological similarity of learned representations [1, 5, 4, 16]. [10] used a shallow, multi-stream network to achieve state-of-the art results in neural predictivity for the mouse visual cortex, showing that similarity in topology impacts similarity in representations. Early work relied on task-driven **loss functions** (e.g., object recognition), but it has recently been demonstrated that models trained using self-supervised loss functions can reach the same or greater level of neural predictivity [24]. Very

4th Workshop on Shared Visual Representations in Human and Machine Visual Intelligence (SVRHM) at the Neural Information Processing Systems (NeurIPS) conference 2022. New Orleans.

few studies [5, 10], however, have investigated the role of the **training data** domain on the neural predictivity of learned representations. While it is thought that the visual stream has specifically evolved around the efficient coding of natural image statistics [18, 11], it is still not clear to what extent this may also apply to DNN models of the brain.

Here, we investigate the role natural image statistics play in the formation of biologically plausible representations within DNN models of the mouse visual cortex. To do this, we first train separate models using self-supervised learning on image datasets from visually distinct domains with varying degrees of "naturalness". We then compare the representations of these networks to each other to confirm whether they are indeed learning distinct feature representations. Finally, we probe the extent to which models trained on domains of varying "naturalness" are predictive of biological neural responses. We find that the naturalness of the training image distribution affects the degree to which the network is able to learn biologically-realistic representations, and find that this is reflected in emergence of canonical visual feature detector motifs such as Gabor filters and textured patches. We see this effect most profoundly in the earlier layers of the artificial network indicating that they are more sensitive to low-level image features. Code and data is available at `https://github.com/talmolab/domain_rep` .

## 2 Methods

### 2.1 Datasets

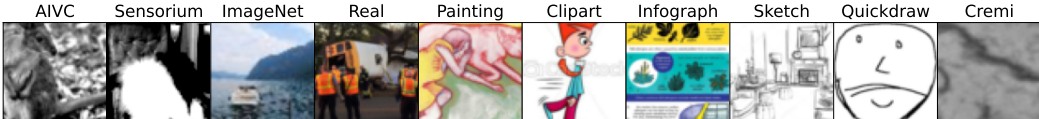

Figure 1: Sample images from each dataset. Networks were trained on ImageNet, DomainNet, and CREMI, then evaluated on AIVC and Sensorium.

**ImageNet** [7] is a multiclass object categorization dataset with $C = 1000$ classes containing a training set with around 1.3 million images and a validation set of around 50 thousand images. In order to control for dataset size and since the dataset is already downsized to $64 \times 64$ (shown to be important for modeling mouse brains [10], we used the TinyImageNet [9] dataset which contains about 100k training images and about 20k validation images with only 200 classes.

**DomainNet** [13] is a large-scale multi-source domain adaptation dataset containing 6 domains each having 345 categories: Real (510 images per category/175,327 total), Infograph (150 images per category/53,201 total), Clipart (150 images per category/48,833 total), Painting (220 images per category/75,759 total), Sketch (220 images per category/70,386 total), and Quickdraw (500 images per category/17,2500 total).

**CREMI**(**C**ircuit **R**econstruction from **E**lectron **M**icroscopy **I**mages) is an electron microscopy dataset made up of three datasets, each consisting of two 5 $\mu m^3$ volumes (training and testing, each $1250 \times 1250 \times 125$ px of serial section EM of the adult fly brain)[1]. We treated each slice in the $z$ dimension as a separate image and then generated 100 $125 \times 125$ tiles per image resulting in a dataset with 27,750 images which we then trained on.

**Sensorium** is a large-scale dataset from mouse primary visual cortex containing the responses of more than 28,000 neurons across seven mice stimulated with thousands of natural images, together with simultaneous behavioral measurements that include running speed, pupil dilation, and eye movements [20].

**Allen Institute Visual Coding Dataset** (**AIVC**) [6, 17] is a mouse neural recording dataset containing electrophysiological and calcium fluorescence recordings of around 60 thousand neurons from the VISp, VISl, VISal, VISam, VISpm, and VISrl regions of 221 different mice. We limited our study to the neural responses of animals in response to the Natural Scenes stimulus set which consisted of

---

[1]MICCAI Challenge on Circuit Reconstruction from Electron Microscopy Images (CREMI): `https://cremi.org`

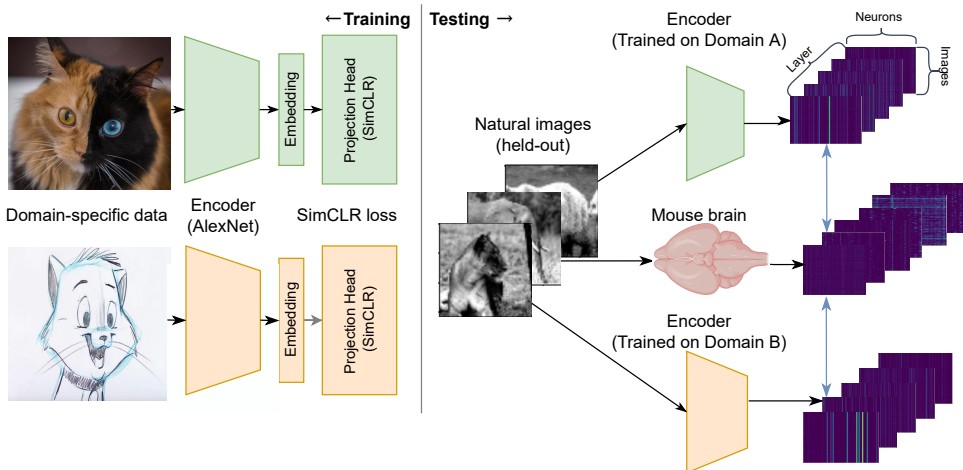

Figure 2: Experiment Pipeline. From left to right: First, we train separate SimCLR-based contrastive AlexNet models on each domain. We then extract activations from each network in response to the AIVC and Sensorium image dataset and evaluate the representational similarity learned by each network to each other as well as to biological neural recordings.

118 grayscale natural images. After following the filtering process outlined by [4] and [10] we were left with around 17,000 neurons across 98 different specimen(calcium) and 1422 neurons across 42 different specimen(neuropixels) across the six visual areas.

## 2.2 Network architecture and training

Because we were interested in only how the input stimuli affected the representations formed, we decided to keep the network architecture fixed and train a network on each domain. Following [10], we chose to use a shallower deep network trained on a contrastive objective function as this has been shown to be the best DNN-based model of the mouse visual cortex. Specifically, we used Pytorch Lightning and the Lightly Self-supervised library to train an AlexNet [8] with a SimCLR loss [3] by replacing the fully-connected layers with a 2-layer MLP projection head that produces a 128-dimensional feature embedding. We trained each network for 600 epochs using the NXent-Loss [3] with temperature scaling $\tau = 0.1$. We used a LARS [23] optimizer with a batch size of 4096, learning rate of 4.8, momentum of 0.9 and weight decay of $10^{-6}$ and we decayed the learning rate by a factor of 0.5 if the loss did not decrease by 0.1 every 10 epochs. To ensure that the distribution of image statistics were not altered due to augmentations, we restricted them to: random-resized crop, random horizontal and vertical flips, and random gray-scaling. Every image dataset was reduced to a resolution of $64 \times 64$ before applying augmentations in order to better model the mouse's visual acuity [14, 10].

## 3 Experiments and Results

### 3.1 Training data domain has an impact on neural predictivity

We first asked whether training contrastive models on different image domains would result in significantly different distributions of activations in response to a (potentially out-of-domain) stimulus set of natural images (AIVC and Sensorium). In other words, we wanted to verify the extent to which DNNs learn the same set of features independent of the data domain. For each network, we extracted the activations from every convolutional layer by forward-propagating each image from the AIVC stimulus set until the activation layer succeeding that convolutional layer to get an $(h \times w \times c)$ tensor.

Given the actions, we calculated the $l_2$-norm of each image activation to measure the distribution of how much each network responded to common images. For each image, we calculated the pairwise Euclidean distance between the activations from each network in order to get an estimate for how far away the representations of the different networks were. Our results (Figure 3) show that the

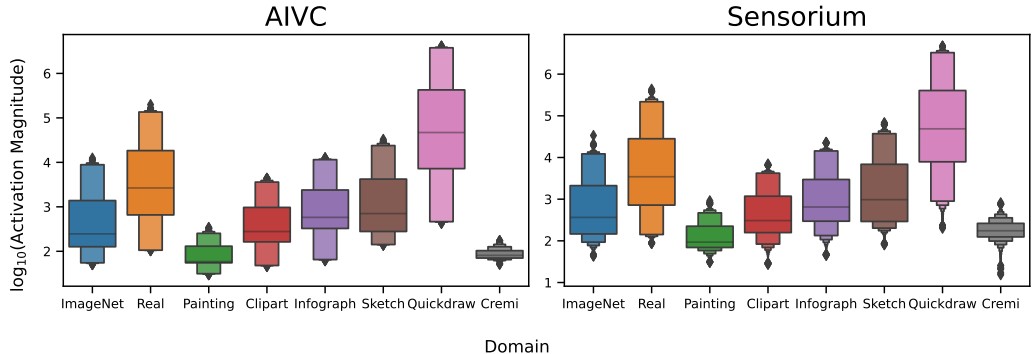

Figure 3: Distributions of activation magnitudes in response to natural images differ across networks trained on different domains. Network activations were extracted per convolutional layer in response to each image in both the AIVC dataset (*left*) and the Sensorium Dataset (*right*)

distribution of activations is indeed different across networks indicating that the networks are learning, unrelated features as a function of data domain.

To verify this result, we hypothesized that if the activations of each network are truly distinct, then it should be possible to find a hyperplane that separates each domain from each other within the activation space. Using the same activations, we trained a separate linear support vector machine (SVM) to classify which domain each set of activations came from. We first flattened the tensors into an $(n \times h \cdot w \cdot c)$ dataset, then performed a 5-fold cross-validation and measured the accuracy, precision, and recall scores for each. Indeed, the SVM was able to classify which domain a given convolutional layer's activation was from with near certainty (classification precision of 0.9876 and 0.9998 for AIVC and Sensorium respectively).

## 3.2 Models trained on different data domains also have different degrees of biological similarity

Given that models trained on different domains exhibit different responses to natural images, we next asked whether these differences in representations were associated with changes in neural predictivity. We calculated the neural predictivity of each layer of each model to each visual region of the AIVC dataset as follows. We first mapped the model activations to their biological counter part on one half of the image set using a partial least squares regression model. We then calculated the inter-animal consistency metric described in [10] using the held out set. However, in our case we decided to keep the measured predictivity per layer rather than for the overall model so we did not take the maximum over all the convolutional layers. We ran 100 trials for each domain using different train-test splits each time.

We find that models trained on different data domains exhibit significant differences in neural predictivity ($p = 1.48 \times 10^{-45}$, $p = 3.46 \times 10^{-58}$, and $p = 2.44 \times 10^{-79}$ using a one-sided Mann-Whitney U test for AIVC-calcium, AIVC-neuropixels, and Sensorium respectively).

## 3.3 Training with more natural image domains increases neural predictivity

We hypothesized that this association may be explained by the "naturalness" of the image domain from which representations were derived. To measure this quantity, we first conducted a spectral analysis on the statistics of a subsample of 1000 images in each dataset to evaluate their adherence to the $\frac{1}{f^p}$ power law that has been previously associated with natural images owing, in part, to their texture bias [19, 18]. We used the distance of the average $p$ value to the idealized $p = 2$ as an approximation for the "naturalness" of the domain.

We find that more natural domains like ImageNet ($p = 2.06 \pm 0.76$) and DomainNet-Real ($p = 1.97 \pm 0.79$) are closer to the previously reported $p = 2$ [19] than less natural domains like DomainNet-Quickdraw ($p = 1.04 \pm 0.31$) and CREMI ($p = 6.19 \pm 0.54$) (Figure 4).

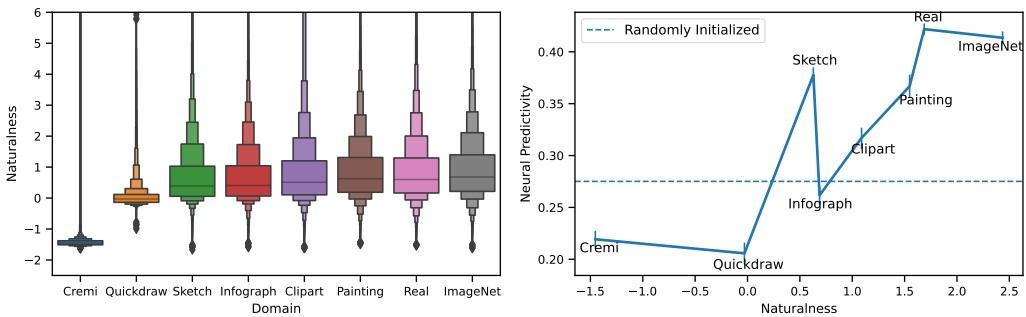

Figure 4: *Left*: Naturalness of each domain approximated by $-\log|p-2|$ after fitting the power spectrum of a sample of images from each domain to the form $\frac{1}{f^p}$. *Right*: The neural predictivity of the first convolutional layer of ANNs trained various domains plotted against its naturalness. The neural predictivity was calculated against the calcium recordings from the AIVC dataset.

Table 1: Neural Predictivity Results.

| Domain | Naturalness | PLS-AIVC(calcium) | PLS-AIVC(ephys) | PLS-Sensorium |
|---|---|---|---|---|
| CREMI | -1.453 | 0.219 | 0.614 | 0.129 |
| Quickdraw | -0.029 | 0.331 | 0.786 | 0.147 |
| Sketch | 0.629 | 0.305 | 0.750 | 0.129 |
| Infograph | 0.688 | 0.278 | 0.755 | 0.143 |
| Clipart | 1.089 | 0.303 | 0.744 | 0.143 |
| Painting | 1.548 | 0.260 | 0.667 | 0.154 |
| Real | 1.687 | 0.342 | 0.808 | 0.156 |
| ImageNet | **2.436** | **0.352** | **0.810** | **0.170** |

Next, using this parameter, we compared the naturalness of models trained on different domains to the neural predictivity of the learned representations. We found that models trained on more natural domains generally appear to exhibit higher neural predictivity (Figure 4, Table 1). This effect is especially clear in early convolutional layers (conv1), with mixed trends for deeper layers deeper (**See Appendix**).

### 3.4 Biological visual system motifs emerge when training with more natural data

Previous work has shown that Gabor-like filter banks emerge in the earlier layers of the visual stream due to the tuning of these neurons to edges and other natural statistics [18], and in computational models of vision [12]. We wondered whether there was a similar functional reason for the gap in neural similarity between naturalistic and non-naturalistic domains. Although qualitative, visualizing the first convolutional kernel of our models suggests that the Gabor-like filters are learned more frequently in models trained on more natural data (**See Appendix**).

## 4 Discussion

Overall, our results show that the input data distribution can have significant impacts on the biological similarity of the representations learned by self-supervised models of the ventral visual stream. Importantly, we demonstrate that the "naturalness" of the data domain has a correlation with the neural predictivity of the learned representations. This effect is most pronounced in the earlier layers of the models, which we hypothesize might be a result of the the earlier layers serving as low level feature detectors (e.g., edge detectors) which would change accordingly with respect to the low level image statistics of the data that it is exposed to. Given that the brain is optimized to natural image statistics, its representations should differ from one optimized for images from other domains. One major limitation of our study, however, is that while our more natural image domains are more "natural" in the spectral sense, they are not natural in the ethological sense. In other words, animals such as mice are less likely to have evolved to learn from images of cars and houses than they are

of shrubbery and predators. Going forward, we believe that it is important not just to take into consideration the architecture and loss function as hyperparameters when modeling the brain with deep neural networks, but also the domain of the training data.

## 5   Acknowledgements

We would like to thank Aran Nayebi and Nathan Kong for their support in replicating their work.

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

# 6 Appendix

## 6.1 Compute

All model training and subsequent analyses were conducted on on-premises GPU computing servers. GPUs were orchestrated through the Run:AI software, which enabled fractional GPU allocations for containerized compute environments. For this work, we used at the most 0.5 of an A40 GPU, the equivalent of 24 GB of GPU RAM. CPU usage was minimal and all analyses downstream of model training were reproduced on Google Colab (CPU-only) and could run on a standard laptop. Our computing environment can be exactly reproduced by using our general-purpose Docker image available at: https://hub.docker.com/r/talmo/tf-extras

## 6.2 Supplementary Figures

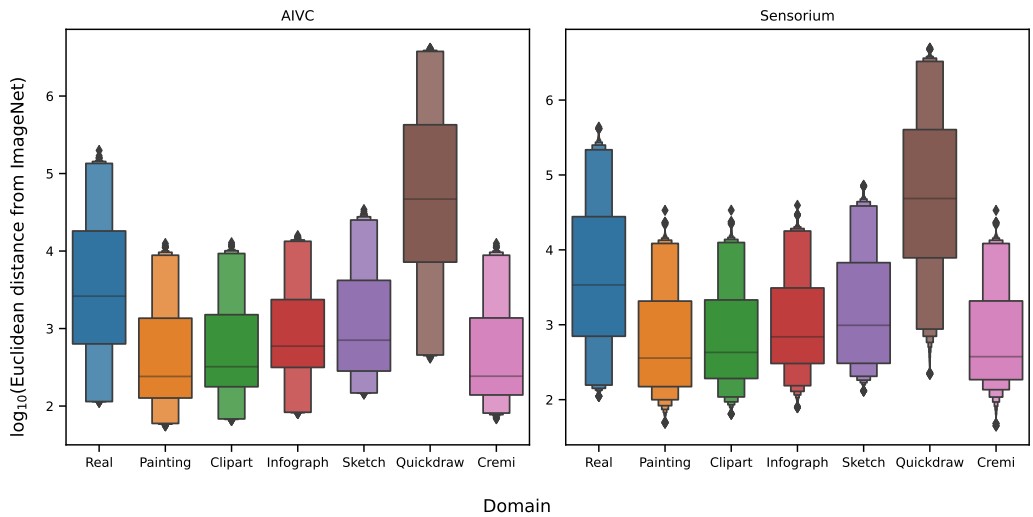

Figure 5: Euclidean distances of activations from each domain from ImageNet

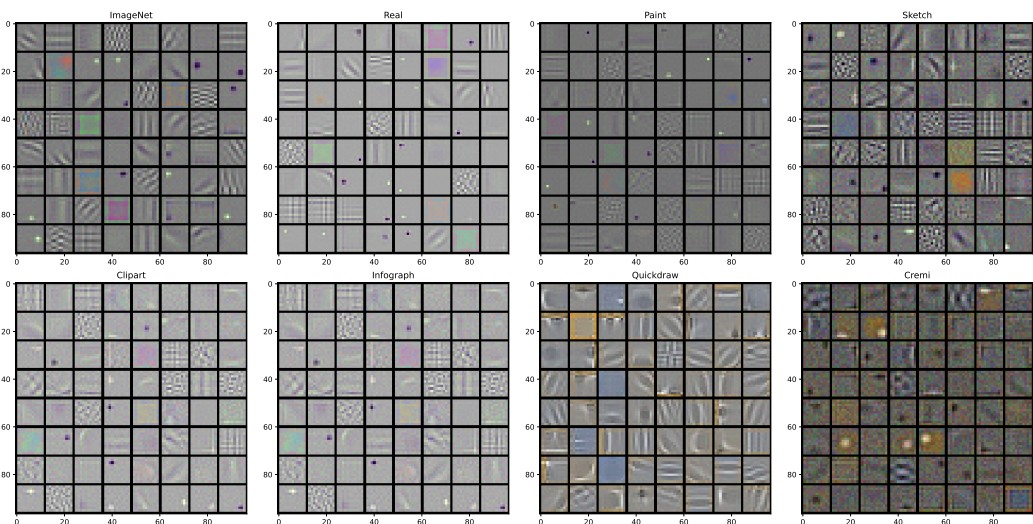

Figure 6: Filters learned by the first convolutional layer of each network

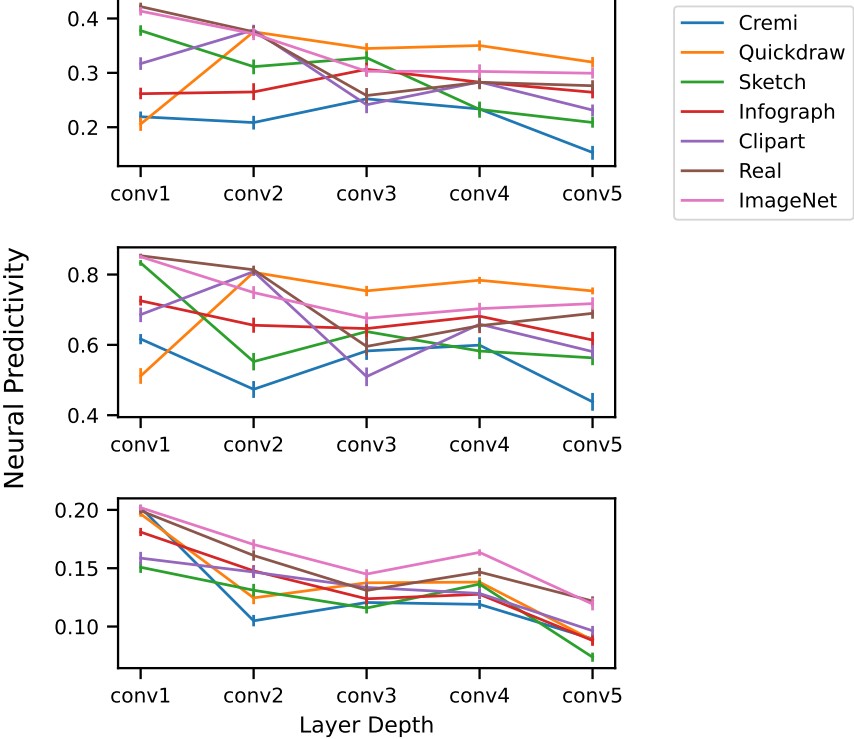

Figure 7: Neural Predictivity as a function of depth. We see a general trend that Neural predictivity decreases in lower layers but we also see that within the first convolutional layer, natural image domains have the higher neural predictivity, while it is less conclusive in later layers. The order is AIVC-calcium, AIVC-neuropixels, and sensorium from top to bottom.

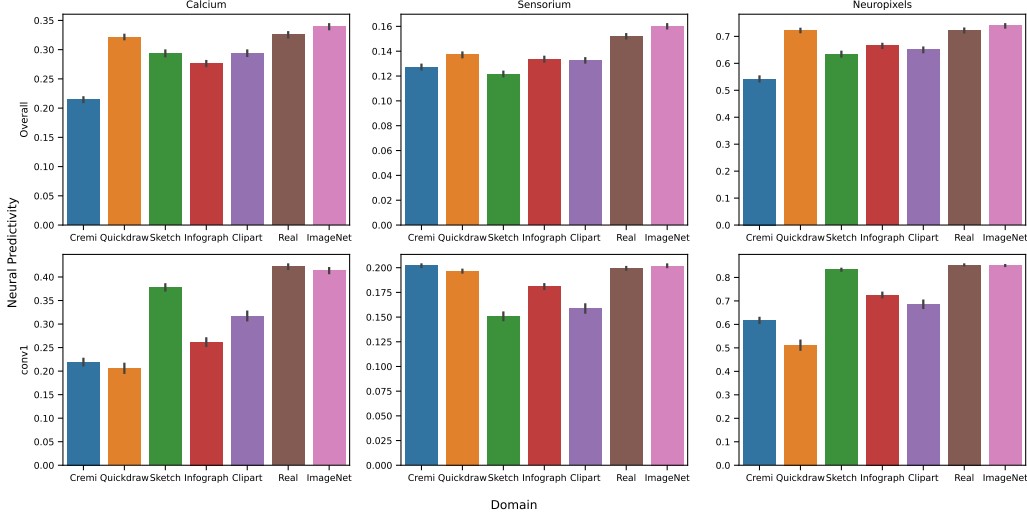

Figure 8: Comparison between the average predictivity over all convolutional layers(top row) and the neural predictivity of conv1 by itself(bottom row). We see that across the board conv1 has higher predictivity. We also see that the more natural domains have a higher level of predictivity

