# OpenReview forum: "Exploring the role of image domain in self-supervised DNN models of rodent brains"
_NeurIPS.cc/2022/Workshop/SVRHM — SVRHM Poster_

### Official Review · Reviewer_h45D · 2022-10-10
**Clear and signifcant and work! Minor issues in figure, table and statistics.**

**Rating:** 8
**Confidence:** 4

**Review:**

It’s a well designed and well conducted study on the effect of training dataset on the representation in the neural networks.

**Clarity:** The motivation, method and result of this paper are crystal clear.

**Significance**: The question it answered filled in a gap in the current computational study of vision. More previous works focused on the effect of network architecture or training objective on the learned representation. Though there are results from this line of work, there is not a systematic effect of architecture etc. This work focused on the orthogonal aspect, i.e. the training dataset for these models and indeed it finds the naturalness of image dataset affects the neural explanability.

This result is intuitive and is consistent with the classic result in neuroscience, that the early visual experience (unsupervised training dataset) of animal will affect their visual cortical representations.

This work will direct more attention to the training datasets of vision models, instead of the training algorithm and architecture.

**Originality**: This work is a clever use case of the self-supervised learning technique (SimCLR) in the last few years. Thanks to these methods, we don’t need labels to learn good visual representations. Thus, any large image sets could be used to train visual representations simply based on a contrastive objective. This enabled the authors to directly compare the visual representation learned on different image datasets and the neural explanability.

I’m sure this paradigm will inspire many following works.

**Pro:**

- The clarity and significance of this paper have been praised above. I really like it! ~ >_< ~

**Cons / Limitations :**

- **Data in Figure 4 right and the Table 1 are inconsistent**.

    The datapoints in Table 1 AIVC column do not correspond to the datapoints in Figure 4. Is there a misuse / misplacement of images?

    In Table 1, caption, it’s not clearly stated which layer the neural predictivity number is based on. Is it conv1 or the whole network?

- **Description of Conv1 filter**

    The Figure 2 in appendix showed the visualizations of conv1 filters in each network, which is intriguing. But I don’t think it could be simply summarized as “*the Gabor-like filters are learned more frequently in models trained on more natural data*”. I see Gabor / grating like features in many of them and the authors didn’t quantify the “frequency”.

    The relationship between conv1 filter and the image statistics is intriguing and should be explored more in the future: I see more curvy kernel in quick draw, I see more bloby kernel in Cremi etc.  In principle, these differences can be grounded in the the statistics in training image patches.

- **Limitation in the “naturalness” statistics.**
One limitation is that the paper’s main claim relies on the notion of **naturalness of an image**. In the paper, this is quantified by the spectral signature of the image and it’s distance to the ideal signature $1/f^2$. This index is well grounded by the classic literature on natural image statistics.

    However, the statistics the authors designed $-\log|p-2|$ didn’t correlate perfectly with neural explanability (Figure 4, Table 1). Moreover **it’s not a very robust statistics**, as when $p$ is close to $2$ a small perturbation will translates to a huge change in the “naturalness” index (Figure 4 Left, large positive naturalness numbers). Is there a better signature of “naturalness” that explains the variance in neural explanability?

    The authors are encouraged to explore more signatures of “naturalness” and see how they connect to the neural explanability.

- **Limitation in the representation analysis**

    If there is a full paper following up, it will be nice to see more analysis on the inner representation of these neural networks, for example: the representation geometry, dimension of the representation, the distribution of prototypes ([Rose, Johnson, Wang, Ponce, Nature Comm. 2021]).

---

### Official Review · Reviewer_avMk · 2022-10-14
**This is a nice paper that explores how training data influences the neural predictivity of learned representations to the mouse visual cortex.**

**Rating:** 7
**Confidence:** 3

**Review:**

In this paper, the authors trained self-supervised neural networks across a range of data domains and evaluated the similarity of the learned representations to the neural activity of the mouse visual cortex. The core conclusion is that the data used to train deep neural networks influences the neural predictivity (brain fit) of the learned representations. This conclusion is reasonable and well supported by the data. In fact, I can’t think of any researchers in the field who would have predicted that training data would not be an important component when modeling the visual system. Since both deep neural networks and biological visual systems are grounded in sensory data, the representations learned in both systems should vary as a function of the training data. As such, the present paper is a useful sanity check: this is the type of result we should expect if the reverse engineering/neuroconnectionism approach was on the right track.

Importantly, this study is one of just a few papers to systematically explore how training data influences the neural predictivity of learned representations. The vast majority of studies explored how architectures and objective functions influence brain fit. As a result, the present study is a welcome addition to the literature.

---

### Official Review · Reviewer_wrbQ · 2022-10-17
**Interesting paper**

**Rating:** 7
**Confidence:** 3

**Review:**

The paper explores biological correlates of self-supervised DNN models trained on various image datasets, and mainly tries to answer the question how robust these correlations are with respect to changes in the "naturalness" of the training images.

Overall, the paper is interesting, clearly written, and seems methodically solid. The results aren't too surprising: the more natural the training data, the better the activations of the models correlate with recordings of mouse neurons.

I wonder why the "sketch" dataset has relatively high correlation, in spite of its low "naturalness". Do the authors have an explanation for this?

---

### Official Review · Reviewer_erDR · 2022-10-17
**Good submission, clarity could be improved**

**Rating:** 7
**Confidence:** 4

**Review:**

This paper studies the influence of training image statistics on the similarity of the representations learnt by neural networks with mouse visual cortex activations. In this work, the authors train several models with images from 10 data sets of various domains (natural images, sketches, paintings, draws, etc.) and study the similarity of the distributions in the learnt activations, as well as the similarity with the activations in mouse visual cortex. The main conclusion is that training image domain has an influence on the activations distribution and the similarity with biological activations.

Overall, I think this is a suitable and good submission for SVRHM. The topic is likely, in my opinion, to be relevant and of interest to the community and the paper provides a broad set of results that shed light on interesting questions.

As aspects that may be improved in the paper, I would point out that the experimental setup could be clearer, as well as the discussion of the results. I have the impression that the authors attempted to fit a large set of results in a short paper and that had an negative impact on the clarity of the document. Specifically, Section 3 could be made clearer. For instance, from the second paragraph in Section 3.1 it is not clear enough to me between what exactly the distances are computed. As another example, Section 3.4 provides very superficial insights and and essentially refers to the appendix for more information.

Furthermore, I would encourage the authors to review some grammar mistakes in the text. For instance, in the abstract:
- "Varying network architectures and loss functions _has_ been shown" -> _have_
- " the highly accurate normative models"
- "each image from the AIVC stimulus set until the activation layer succeeding that convolutional layer to get an (h × w × c) tensor"

Other comments:

- In Section 2, the authors write that they "chose to use a shallow network". Then, they specify they chose AlexNet, which is not a shallow network. Could you clarify?
- In Section 3.1, the authors study whether the activations of models trained with different data domains can be classified into the corresponding domain, to conclude that if possible, then the activations are "truly distinct". However, technically the activations may be perfectly classified by a machine learning algorithm without being fundamentally different.